# Cerebral Amyloid Angiopathy Related Inflammation: A Single-Center Case Series Analysis

**DOI:** 10.3390/brainsci15050472

**Published:** 2025-04-29

**Authors:** Syed Zahid Ali, Hanah Alley, James Johnson, Harshini Sirvisetty, Michael Sowell, Alex Glynn, Peter Hedera

**Affiliations:** 1Department of Neurology, School of Medicine, University of Louisville, 500 S Preston St., Louisville, KY 40202, USA; hanah.alley@louisville.edu (H.A.); michael.sowell@louisville.edu (M.S.); peter.hedera@louisville.edu (P.H.); 2School of Medicine, University of Louisville, 323 E Chestnut St., Louisville, KY 40202, USA; harshini.sirvisetty@louisville.edu; 3Kornhauser Health Sciences Library, University of Louisville, 540 S Preston St., Louisville, KY 40202, USA; alex.glynn@louisville.edu

**Keywords:** cerebral amyloid angiopathy-related inflammation, neuroimaging, cerebral microbleeds, amyloid angiopathy

## Abstract

Background: Cerebral amyloid angiopathy-related inflammation (CAA-RI) is a rare subtype of cerebral amyloid angiopathy (CAA), which presents mostly as a subacute and reversible encephalopathy. Primary symptoms include behavioral changes and cognitive decline in the form of rapidly progressive dementia, headache, seizures, and focal neurological deficits. It can also manifest as a varied range of typical and atypical presentations. Misdiagnosis is common because it shares symptoms with other infectious, ischemic and autoimmune pathologies and there is also a significant overlap of MRI findings. Methods: Gold standard diagnosis requires brain biopsy in appropriate clinical setting, but diagnostic criteria is established for probable and possible CAA-RI using clinical symptoms and MRI findings in the absence of other inflammatory, infectious or autoimmune processes. Immunomodulatory therapy is the mainstay of treatment, with variable response. Results: We present a case series of three patients with CAA-RI highlighting disease course, neuroradiological manifestation, treatment response, and clinical outcomes. We also provide a literature review to increase insight into this rare pathology. Conclusions: Early diagnosis and prompt initiation of immunosuppressive therapy is beneficial in most cases.

## 1. Background and Rationale

Cerebral Amyloid Angiopathy-related inflammation (CAA-RI) is a less common, inflammatory variant of Cerebral Amyloid Angiopathy (CAA). CAA-RI is a reversible encephalopathy syndrome most commonly presenting with subacute encephalopathy and complaints of cognitive decline along with headache, seizure, and focal neurologic deficits [1,2]. The definitive diagnosis of CAA-RI requires pathology-confirmed brain biopsy. However, in recent years, criteria only requiring clinical and radiographic evidence has been validated to diagnose either probable or possible CAA-RI without biopsy [3]. Prompt recognition and diagnosis is essential as patients’ symptoms due to CAA-RI tend to respond to immunosuppressive agents, most commonly intravenous high-dose corticosteroid pulse with oral taper of steroids [4,5,6].

CAA-RI is described in the literature as most commonly being monophasic but relapsing courses have been described and are likely more common than previously documented [7,8]. Relapsing symptoms tend to occur when immunosuppression is discontinued or tapered [4,7,9].

We present three cases of probable CAA-RI who were admitted to our hospital with various symptoms. Two of these patients received IV steroids followed by oral taper and had significant improvement in symptoms. One patient had mild symptoms at presentation and did not require any immunomodulatory therapy.

## 2. Case Summaries

### 2.1. Patient 1

A 75-year-old female with a history of hypertension, hyperlipidemia, diabetes mellitus type 2, hypothyroidism and fibromyalgia presented as a transfer from an outside hospital with word finding progressive difficulty and cognitive decline. Her MRI showed diffuse edema scattered more prominently in the left cerebral hemisphere, confluent bilateral white matter hyperintensities, and susceptibility-weighted imaging concerning amyloid angiopathy (Figure 1a,b). Lumbar puncture was performed, and showed normal cell count and metabolites without any evidence of infection or autoimmune process. She met diagnostic criteria for probable CAA-RI and was treated with high-dose oral dexamethasone. She initially responded to this therapy with subjective improvement in cognition. During the tapering schedule of dexamethasone, she deteriorated, with worsening of her cognitive dysfunction six weeks after starting initial treatment. On repeat imaging, she was found to have had an acute stroke. The patient started on antithrombotic and statin for secondary prevention. Her mental status failed to improve. She was subsequently treated with another course of high-dose intravenous methylprednisolone, and she had significant improvement in cognitive symptoms on day 4 of treatment. Formal neuropsychiatric evaluation was performed five months post-steroid-pulse therapy and significant improvement in cognitive function was noted. At the follow-up appointment one-year post-steroid pulse (about 17 months after discharge), she continues to endorse cognitive improvement in performing independent activities of daily living.

### 2.2. Patient 2

Patient is a 46-year-old male without any significant past medical history who presented as a transfer from outside hospital after having a motor vehicle accident secondary to a prolonged seizure while driving. His imaging showed a left occipital hemorrhage and confluent symmetric bilateral white matter hyperintensities (Figure 1c,d). Subsequent MRI confirmed hemorrhage on GRE sequence but also showed multiple areas of lobar cerebral microbleeds. EEG demonstrated polymorphic background seen over both posterior head regions, worse on the left, suggestive of a structural lesion in the left hemisphere. Lumbar puncture was performed, and did not show any obvious abnormality. He was started on seizure prophylaxis. He did not receive any steroids as his seizures were controlled and his clinical status improved rapidly.

### 2.3. Patient 3

A 72-year-old male with prior medical history significant for cerebral amyloid angiopathy, hypertension, hyperlipidemia and prostate cancer treated with prostatectomy, presented with confusion, and intermittent periods of staring and lip smacking. Patient was initially started on anti-seizure medications because of suspicion for epileptiform events. EEG monitoring was consistent with non-epileptic events and showed diffuse dysfunction without active seizure-like activity. He remained severely altered and encephalopathic and his level of consciousness declined. His MRI showed areas of gliosis, encephalomalacia, confluent white matter hyperintensities and numerous bilateral cerebral microbleeds (Figure 1e,f). Lumbar puncture did not show any sign of infectious or autoimmune processes. He underwent PLEX therapy followed by IVIG therapy because of suspicion for an autoimmune encephalitis. He did not improve, so the diagnosis of probable CAA-RI was also considered. He was treated with five days’ course of high-dose steroids. His condition improved with high-dose IV steroids, followed by oral steroids taper. Goals of care were changed by the family, and referral to an outpatient palliative care program was placed.

Bloodwork and cerebrospinal fluid analysis for all three patients are provided in Table 1. Timelines of each patient’s clinical course are provided in Figure 2.

## 3. Discussion

Cerebral amyloid angiopathy-related inflammation is a rare disorder characterized by inflammatory reaction to deposits of Aβ (amyloid-β) in cerebral blood vessels, specifically in tunica media and adventitia [10]. It commonly occurs in elderly patients. Typical clinical features include cognitive impairment, stroke-like focal neurological deficits, seizures, altered mental status, and headache. Cognitive impairment is usually mild but, in some cases, it can be rapidly progressive with severe mental decline occurring in a matter of days to weeks [1].

Diagnosis of CAA-RI is challenging as there are no definite differences in neuroimaging findings seen in CAA and CAA-RI. Frequent neuroimaging findings in CAA include multiple lobar cerebral microbleeds (CMBs), cortical superficial siderosis (CSS) or subarachnoid hemorrhage (SAH), white matter hyperintensities (WMHs), enlarged perivascular spaces (EPVs) commonly in the centrum semiovale (CSO), and cortical atrophy. All these neuroimaging abnormalities can be also seen in CAA-RI. Evidence of inflammation solely based on neuroimaging is very difficult to establish [11,12].

Thus, original diagnostic criteria mandate neuropathological evidence of inflammation [13]. Considering the risk-to-benefit ratio, performing biopsy in an elderly population, in which brain parenchyma is prone to hemorrhages, is a difficult decision for patients and physicians. Due to increased recognition of CAA-RI cases, and extensive review of neuroimaging findings, it is now feasible to manage these cases with increased diagnostic certainty, while avoiding the risks of brain biopsy [3]. When biopsy is not a safe consideration, in the right clinical setting, MRI is extremely helpful in diagnosing CAA-RI cases and monitoring treatment response.

As mentioned earlier, most frequent neuroimaging findings in cases of CAA-RI include widespread CMBs, WMH, EPVs, and CSS. CSF analysis may help in the right clinical settings to rule out other infectious, autoimmune, or paraneoplastic condition, which can present in a similar fashion [14,15].

None of our patients underwent biopsy for neuropathological examination, but all of them met the diagnostic criteria for probable CAA-RI (Table 2). Patients 1 and 3 received IV steroids followed by oral taper with significant improvement of symptoms. Patient 2 was relatively young and had no known medical history. There was no family history of dementia, and his symptoms improved even without use of anti-inflammatory agents.

Other options for immunosuppression are also available, but the majority of cases respond very well to readily available intravenous and oral corticosteroids. Therefore, these options are reserved only for those cases which do not show improvement. These options include methotrexate, mycophenolate–mofetil, cyclophosphamide, immunoglobulins and azathioprine [16,17,18].

Our reported cases demonstrate the importance of early diagnosis of CAA-RI based on a high index of suspicion, and further support administration of high-dose steroids as an effective treatment for this condition.

The limitations of this case series include the small number of patients and length of follow-up. While one patient was followed up to 17 months post-discharge, the other two patients have been out of the hospital less than 6 months at time of writing. A longer observation period will be necessary to characterize recovery patterns and treatment efficacy. While a definitive diagnosis of CAA-RI would require histopathology, given its invasive nature, it is generally avoided and not preferred by patients or families; this occurred with all patients in our case series analysis.

## 4. Conclusions

Our personal experience of diagnosis and treatment of a case of CAA-RI and extensive review of the available literature indicate that lobar CMBS and white matter hyperintensities are the most frequent neuroimaging findings in CAA-RI. The most frequent symptom is cognitive impairment and behavioral changes, but new onset seizure can be the first symptom to present. Diagnosis is challenging. If untreated, disease can be aggressive and fatal. Immunosuppression is the mainstay of treatment, and treatment response varies in different patients. Early diagnosis and prompt initiation of immunosuppressive therapy is beneficial in most cases. Treatment becomes more challenging in cases of other concomitant pathologies, like stroke, epilepsy, systemic and central nervous system infections. We recommend close follow-up of these patients after treatment to see long-term complications and monitor cognitive decline. Future research projects should include extensive search for preventable risk factors for this condition if any exist. There is also a dire need to look for early serum biomarkers in CAA-RI which can raise clinical suspicion, and aid in diagnosis and monitoring disease course and treatment response.

## Figures and Tables

**Figure 1 brainsci-15-00472-f001:**
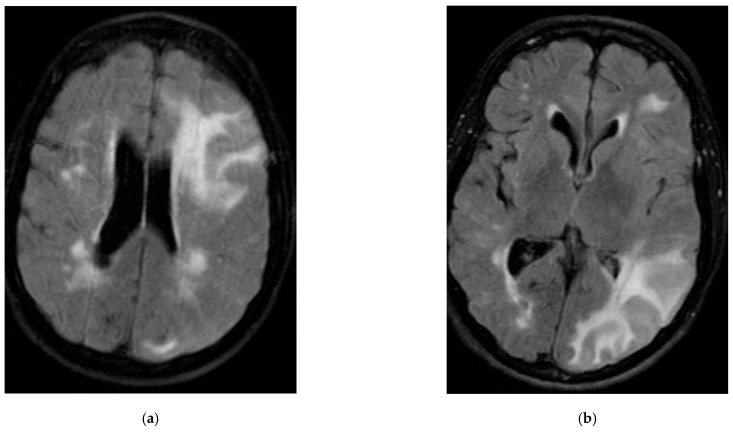
Imaging abnormalities associated with CAA-RI: (**a**) Patient 1: extensive T2-weighted-fluid-attenuated inversion recovery (FLAIR), patchy and confluent hyperintensities, most pronounced in left frontal parietal area. (**b**) Patient 1: extensive T2 FLAIR, confluent hyperintensities in right parieto-occipital area. (**c**) Patient 1: peripheral foci of gradient echo blooming, widely distributed and diffuse, indicative of cerebral microbleeds. (**d**) Patient 1: numerous peripheral foci of gradient echo blooming, suggestive of cerebral microbleeds. (**e**) Patient 2: punctate foci of gradient echo low signal, suggesting scattered cerebral microbleeds. (**f**) Patient 2: foci of gradient echo low signal, suggestive of cerebral microbleeds. (**g**) Patient 2: an area of intraparenchymal in the left occipital region, measuring approximately 1.4 cm. (**h**) Patient 3: numerous scattered foci of signal voids on gradient recalled echo (GRE), suggestive of cerebral microbleeds. (**i**) Patient 3: areas of gradient echo low signal, showing a predominant peripheral distribution, mostly pronounced in right occipital lobe. (**j**) Patient 3: areas of encephalomalacia and gliosis in right precuneus and right middle frontal gyrus, suggestive of sequel of lobar hemorrhages.

**Figure 2 brainsci-15-00472-f002:**
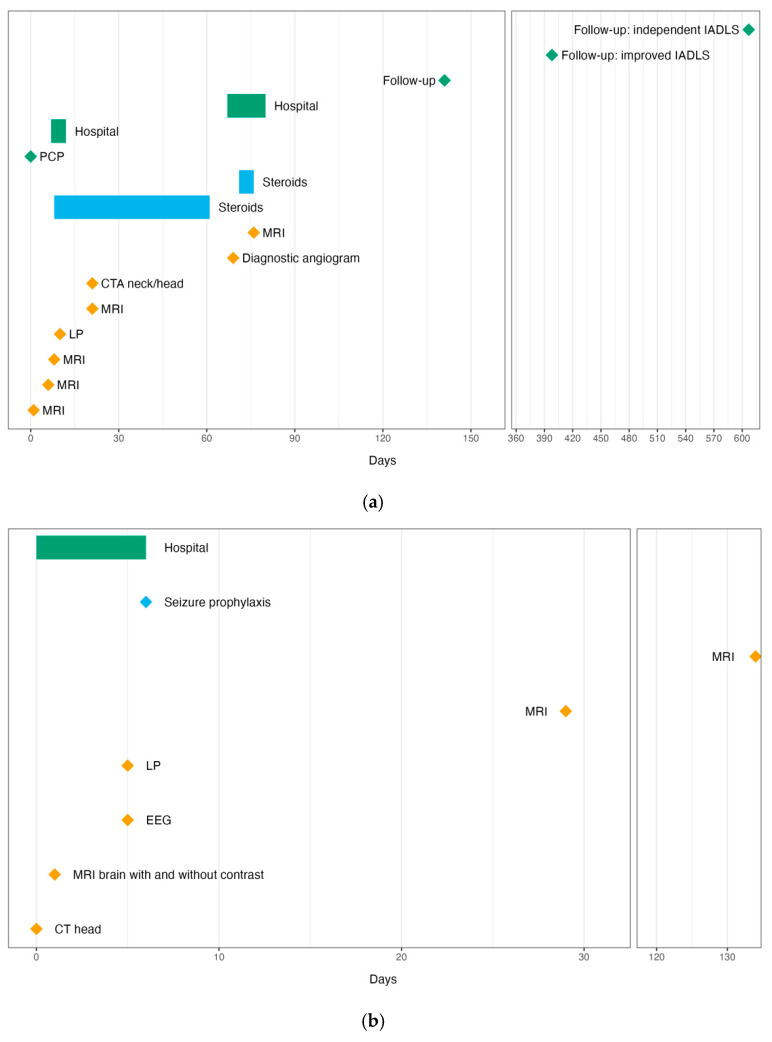
Clinical courses of all three patients: (**a**) Patient 1; (**b**) Patient 2; (**c**) Patient 3. CT, computed tomography; CTA, computed tomography angiography; EEG, electroencephalogram; IADLS, instrumental activities of daily living; IVIG, intravenous immunoglobulin; LP, lumbar puncture; MRI, magnetic resonance imaging; PLEX, plasmapheresis.

**Table 1 brainsci-15-00472-t001:** Basic metabolic panel, complete blood count, and cerebrospinal fluid analysis results.

	Patient 1	Patient 2	Patient 3
BMP			
Sodium	140	134	143
Potassium	4.3	3.3	2.9
Chloride	107	103	98
CO_2_	23	23	31
Anion gap	10	8	14
Glucose	154	99	94
BUN	18	14	14
Creatinine	0.58	1.04	1.17
GFR	94.5	90	66
BUN/creatinine ratio	31	13.5	12
Calcium	9.3	8.9	9.2
CBC			
WBC	21.5	6.5	8.4
RBC	4.27	5.23	4.77
Hemoglobin	12.8	15.8	14.2
Hematocrit	40.2	44.6	40.5
MCV	94.1	85.2	84.9
MCH	30	30.2	29.7
MCHC	31.9	35.5	35
Platelets	477	193	240
MPV	7	9.3	8
RDW	13.1	14.6	14.1
CSF			
Total nucleated cells	0	1	44
Total RBCs	1	40	8501
Neutrophils	0	9	70
Lymphocytes	0	41	16
Monocytes	0	50	14
Glucose	125	62	72
Protein	29.7	34	91
Mayo Clinic ENC2	Negative	Negative	Negative

BMP, basic metabolic panel; BUN, blood urea nitrogen; CBC, complete blood count; ENC2, encephalitis panel; GFR, glomerular filtration rate; MCH, mean corpuscular hemoglobin; MCHC, mean corpuscular hemoglobin concentration; MCV, mean corpuscular volume; MPV, mean platelet volume; RBC, red blood cell count; RDW, red blood cell distribution width; WBC, white blood cell count.

**Table 2 brainsci-15-00472-t002:** Criteria to diagnose CAA related inflammation. Adapted from Chung [13] and Auriel et al. [3].

Grade of Probability	Criteria
Possible CAA-RI	Age ≥ 40 years. More than one of the following symptoms not directly attributable to an acute ICH: 1. Headache 2. Impaired consciousness 3. Behavioral change 4. Focal neurological deficit 5. Epileptic seizures MRI with WMH lesions that extend only to neighboring subcortical white matter. More than one of the following cortico-subcortical hemorrhagic lesions: 1. Cerebral macrobleeds 2. Cerebral microbleeds 3. Cortical superficial siderosis Absence of other infectious or neoplastic causes.
Probable CAA-RI	Age ≥ 40 years. More than one of the following symptoms not directly attributable to an acute ICH: 1. Headache 2. impaired consciousness 3. behavioral change 4. Focal neurological deficit 5. Epileptic seizures MRI: asymmetric, uni- or multifocal WMH-lesions in the proximate subcortical white matter. Asymmetry is not in setting of previous ICH. More than one of the following cortico-subcortical hemorrhagic lesions: 1. Cerebral macrobleeds 2. Cerebral microbleeds 3. Cortical superficial siderosis Absence of other infectious or neoplastic causes.
Definite CAA-RI	Criteria of probable CAA-RI plus histopathology findings:Perivascular, transmural and/or intramural inflammation. Proof of amyloid deposits in vessels of affected cortex and leptomeningeal regions.

## Data Availability

The original contributions presented in this study are included in the article. Further inquiries can be directed to the corresponding author.

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
