# Peer review of "Cerebral Amyloid Angiopathy Related Inflammation: A Single-Center Case Series Analysis"

_brainsci, 2025, doi:10.3390/brainsci15050472_

Round 1

Reviewer 1 Report

Comments and Suggestions for Authors

«Cerebral amyloid angiopathy related inflammation: A case series and literature review».

This article focuses on the insufficiently researched and relatively uncommon condition of cerebral amyloid angiopathy-related inflammation (CAA-RI). The diagnostic challenges of CAA-RI— including nonspecific neurological symptoms that overlap with other disorders and the lack of definitive diagnostic tools—make this study particularly valuable for practicing physicians and neuroscientists. The authors emphasize the potential of non-invasive diagnostic approaches, such as MRI, which are less invasive and risky than biopsy. They consistently highlight the importance of developing early detection methods and initiation of immunosuppressive therapy which could significantly improve the effectiveness of medical treatment. This comprehensive review provides relevant and insightful information, creating an overall positive impression.

This work has several strong points:

  1. The presented case reports effectively illustrate the diversity of symptoms and clinical courses of CAA-RI, along with diagnostic methodologies. Each patient’s medical history is described in a clear, systematic manner, supplemented with supporting images that illustrate the findings.
  2. A comprehensive literature review is provided, with a dedicated section summarizing diagnostic criteria for the disease. This structured approach facilitates a deeper understanding of the study’s key topic.
  3. The article is well-written and accessible, free of grammatical or typographical errors. The structure is logical and includes all relevant sections.

I can highlight only two limitations of this study:

  1. While the authors’ focus on non-invasive diagnostics and avoiding potentially traumatic procedures is commendable, it must be acknowledged that even with highly accurate diagnostic criteria and successful therapeutic outcomes, the absence of histopathological confirmation reduces diagnostic certainty in these cases.
  2. The longest follow-up period post-treatment was only one year—and for just one patient. Extended observation would provide a more comprehensive understanding of recovery patterns and treatment efficacy.

These considerations do not diminish the overall quality of the article or the value of the authors' work. The study will prove valuable for practicing clinicians due to its clear presentation of diagnostic criteria and therapeutic approaches. The authors have successfully integrated their own clinical experience with comprehensive literature data, making the article both informative and relevant. For future research, it would be advisable to include a larger patient cohort and conduct longer-term follow-up observations. Overall, I recommend this article for publication.

Author Response

Comment 1: While the authors’ focus on non-invasive diagnostics and avoiding potentially traumatic procedures is commendable, it must be acknowledged that even with highly accurate diagnostic criteria and successful therapeutic outcomes, the absence of histopathological confirmation reduces diagnostic certainty in these cases.

Response 1: We agree and have added a note about this limitation at the end of the Discussion section.

Comment 2: The longest follow-up period post-treatment was only one year—and for just one patient. Extended observation would provide a more comprehensive understanding of recovery patterns and treatment efficacy.

Response 2: The follow-up period for patient 1 was almost a year and a half; we have added wording in the case summary to make this clearer. However, the point is well taken, and we have added a note about this limitation at the end of the Discussion section.

Comment 3: For future research, it would be advisable to include a larger patient cohort and conduct longer-term follow-up observations. Overall, I recommend this article for publication.

Response 3: We have acknowledged these limitations at the end of the Discussion section.

Reviewer 2 Report

Comments and Suggestions for Authors

Ali et al. present an interesting and valuable case series on cerebral amyloid angiopathy-related inflammation (CAA-ri). The scientific relevance of the report is unquestionable, as it contributes to the growing body of literature on this under-recognized and often challenging condition. However, several aspects of the manuscript could be strengthened to improve clarity and completeness:

  1. The current title, "Cerebral amyloid angiopathy related inflammation: A case series and literature review," does not accurately reflect the manuscript’s focus. While the manuscript includes a contextual review of prior studies, its core contribution lies in presenting three well-documented cases from a single center. A more appropriate title might be:
    "Cerebral amyloid angiopathy-related inflammation: A single-center case series analysis."

  2. Add  table summarizing the diagnostic workup performed in each case (including CSF analysis, blood exams) would be extremely helpful for readers to compare and synthesize the clinical data across the cases.

  3. Include a timeline figure illustrating the clinical course of each patient—from symptom onset to diagnosis, treatment, and follow-up—would enhance understanding of the disease trajectory and management decisions.

  4. It would be beneficial to include additional MRI images for each case, specifically T2, FLAIR, and Gradient Echo/SWI sequences. For each sequence, please annotate or describe the key radiological findings that support the diagnosis of CAA and CAA-ri in the image itself. This is particularly important given that many readers may not be familiar with the typical imaging features of this condition.

Author Response

Comment 1: The current title, "Cerebral amyloid angiopathy related inflammation: A case series and literature review," does not accurately reflect the manuscript’s focus. While the manuscript includes a contextual review of prior studies, its core contribution lies in presenting three well-documented cases from a single center. A more appropriate title might be: "Cerebral amyloid angiopathy-related inflammation: A single-center case series analysis."

Response 1: We agree that this title is more appropriate and have changed it accordingly.

Comment 2: Add table summarizing the diagnostic workup performed in each case (including CSF analysis, blood exams) would be extremely helpful for readers to compare and synthesize the clinical data across the cases.

Response 2: We have added the table. As documented in the text, lumbar punctures for all three patients showed no significant findings.

Comment 3: include a timeline figure illustrating the clinical course of each patient—from symptom onset to diagnosis, treatment, and follow-up—would enhance understanding of the disease trajectory and management decisions.

Response 3: we have added this as figure 2.

Comment 4: It would be beneficial to include additional MRI images for each case, specifically T2, FLAIR, and Gradient Echo/SWI sequences. For each sequence, please annotate or describe the key radiological findings that support the diagnosis of CAA and CAA-ri in the image itself. This is particularly important given that many readers may not be familiar with the typical imaging features of this condition.

Response 4: We expanded Figure 1 with more images and written descriptions as requested.
